# Association between Contextual Factors and Affordances in the Home Environment of Infants Exposed to Poverty

**DOI:** 10.3390/children10121932

**Published:** 2023-12-15

**Authors:** Carolina Fioroni Ribeiro da Silva, Ana Luiza Righetto Greco, Denise Castilho Cabrera Santos, Giuseppina Sgandurra, Eloisa Tudella

**Affiliations:** 1Neuropediatrics and Motricity Studies Center (NENEM), Federal University of São Carlos, São Carlos 13565-905, Brazil; etudella@ufscar.br; 2Department of Physiotherapy, Federal University of Uberlandia, Uberlandia 38405-314, Brazil; anagreco@ufu.br; 3Human Movement Sciences Graduate Program, Methodist University of Piracicaba, Piracicaba 13423-170, Brazil; dcsantos@unicamp.br; 4Department of Human Development and Rehabilitation, School of Medical Science, University of Campinas (UNICAMP), Campinas 13083-970, Brazil; 5Department of Clinical and Experimental Medicine, University of Pisa, Via Roma 67, 56126 Pisa, Italy; giuseppina.sgandurra@fsm.unipi.it; 6Department of Developmental Neuroscience, IRCCS Fondazione Stella Maris, Viale del Tirreno 331, 56128 Pisa, Italy

**Keywords:** maternal age, physical space, variety of stimulation, home environment, toys

## Abstract

Home environments of infants exposed to poverty exhibit fewer affordances for child development. This study aimed to investigate the association between contextual factors and affordances in the home environments of infants facing poverty. Term infants (*n* = 128) were divided into two groups: (1) exposed group (EG), comprising term infants exposed to poverty, and (2) comparison group (CG), consisting of term infants classified with medium and high socio-economic status. The main dependent variables were physical space, variety of stimulation, gross and fine motor toys, and the total score; measured using the Brazilian version of the Affordances in the Home Environment for Motor Development-Infant Scale (AHEMD-IS) questionnaire, named *Affordances no Ambiente Domiciliar para o Desenvolvimento Motor-Escala Bebê*. Statistical analysis employed independent sample t-tests or Mann–Whitney tests, chi-square tests, and stepwise multiple linear regression models to evaluate predictors of less adequate home environments. The EG presented significantly fewer affordances in gross motor toys (Cohen’s r = 0.353; *p* < 0.01); fine motor toys (Cohen’s r = 0.327; *p* < 0.01); and total score (Cohen’s r = 0.377; *p* < 0.01). Binary logistic regression analysis showed a significant association (r^2^ = 0.828, *p* < 0.01) between the less than adequate home environment category and maternal age (*p* = 0.043, OR: 0.829 (0.692–0.994)), revealing that maternal age was associated with better quantity and quality of affordances for child development.

## 1. Introduction

Child development is a complex, dynamic, and multifactorial process influenced by biological, psychological, and environmental factors. In this perspective, the approach to infant care must be biopsychosocial [1]. According to the biopsychosocial model, risk factors may act as barriers, while protective factors may act as facilitators for functionality. Therefore, they should be considered in the assessment and monitoring of child development [2]. Biological and/or environmental risks are defined as risk factors that increase the probability of deficits in child development. In other words, these factors may encompass both biological and psychosocial dangers and/or threats that infants encounter during child development. Conversely, protective factors play a crucial role in promoting resilience, allowing infants to overcome adversities [3]. For example, high socioeconomic and cultural conditions, daily practices, maternal education, the physical space in which infants live, and the variety of toys and stimulation offered to them may present protection for child development. These factors significantly influence functionality and the quality of life of infants and their families [4,5,6]. Poverty is a concerning environmental factor worldwide. In developing countries, it has been observed that 1.3 billion people live in multidimensional poverty [7].

Healthy infants who do not receive sufficient opportunities to engage in activities may encounter greater difficulties in realizing their developmental potential, such as infants exposed to multiple environmental risk factors [8,9]. The analysis of the opportunities for action offered by the environment for infant child development aligns with the concept of affordance. Affordance refers to the possibilities of action or movement offered by the environment to an infant [10]. It includes the ability to perceive and respond to elements such as objects, events, and interaction with others in the environment. For example, before reaching for an object, infants assess its size and decide whether to reach for it using one or two hands [10]. To enable effective performance of actions in the environment, it is important that the affordances offered by objects are adequate, and that the infant can perceive the necessary characteristics for functional action [11].

The affordances within the home environment play an important role in relation to child development [12]. Therefore, a comprehensive evaluation of the home environment in the initial months of life becomes essential, encompassing quantitative and qualitative assessments across different dimensions, such as physical space, variety of stimulation, and the availability of gross and fine motor toys [13,14]. Freitas et al. [15] also found that infants aged between 3 and 18 months, exposed to poverty, lack sufficient opportunities for developmental activities within their environment. On the other hand, infants aged 6 to 9 months from families with a high socioeconomic status (SES) showed better performance in motor, cognitive, and language development due to the greater availability of opportunities for action [16]. Recently, studies found that the higher the SES and number of residents, the greater the affordances in the home environment of children from 8 months [17]. However, the authors assessed only the frequencies of affordances at home. We do not know how these factors may predict adequate affordances for child development in the home environments of younger infants and how the characteristics of the contextual factors may influence the affordances of infants exposed to a low SES.

In addition, the poverty effect in the home environment is harmful regardless of the ethnic group [18], as observed by the Home Observation for Measurement of the Environment-Short Form [19], a tool utilized in the United States and not translated for Brazilian Portuguese for early age. Based on this context, in this study, we looked for instruments to assess the home environment which were translated for upper–middle income countries, such as the Brazilian version of the Affordances in the Home Environment for Motor Development-Infant Scale (AHEMD-IS) questionnaire, named Affordances *no Ambiente Domiciliar para o Desenvolvimento Motor-Escala Bebê* [13,14].

Based on previous studies, this study is a step forward in understanding the impact of environmental factors on providing adequate affordances for early childhood development. The findings of this study seek to explore and optimize possible protective factors that could bring benefits to infants exposed to poverty.

The objectives of this study were to address the following inquiries: (1) In which dimensions of the home environment do infants exposed to poverty encounter fewer affordances for child development when compared to the home environments of comparable infants who are not exposed to poverty? (2) Are there contextual factors associated with the quantity and quality of affordances that are less than adequate for child development?

Based on the aforementioned context, we hypothesized that: (1) the home environments of infants exposed to poverty present less affordances for child development when compared to that of infants not exposed to poverty; (2) that high SES, the number of children living in the household, and maternal education and/or age may be factors associated with the provision of adequate affordances for child development. It is assumed that siblings play a stimulating and challenging role, encouraging infants to engage in playful movement within the space, and more experienced mothers with high SES have more access to knowledge, thereby providing better affordances for child development.

## 2. Materials and Methods

### 2.1. Design

This is an observational, cross-sectional, prospective study, employing convenience sampling. The study adhered to the guidelines outlined in the Strengthening the Reporting of Observational Studies in Epidemiology [20].

### 2.2. Ethical Procedures

The study was approved under opinion number 3,203,794 and CAAE: 04097718.9.0000.5504, by the Ethics Committee for Research on Human Beings of Federal University of São Carlos, in accordance with the Guidelines and Regulations for Research Involving Human Beings (Resolution 466/2012, from the National Health Council).

### 2.3. Participants

The ideal sample size is 5 to 10 times the number of independent variables [21]. Term infants [22] of both sexes, aged 3 and 4 months, and classified as low SES according to the poverty income ratio (PIR), were chosen for the inclusion in the exposed group (EG); in contrast, term infants of both sexes, aged 3 and 4 months, and not classified as having a low SES, were selected for the comparison group (CG). Infants only participated in this study if their caregivers agreed to participate in the research by signing the informed consent form. Infants with pre- (e.g., intrauterine growth restriction), peri- (e.g., anoxia, hypoxia, Apgar < 7, low birth weight), or post-natal (e.g., neurological impairment, auditory, or visual deficits) complications were not included in this study, including those with sensorial, genetic syndromes, musculoskeletal, cardiac alterations, as diagnosed or reported by the caregiver.

### 2.4. Instruments

Adapted sociodemographic questionnaire: this questionnaire encompasses items related to anamnesis, birth data, anthropometric measurements (such as, weight, length, and head circumference), posteroanterior and binaural features, thoracic details, medical history, life history, and the characteristics of the family context [23].

For the assessment of poverty, SES was measured using the Poverty Income Ratio (PIR), which uses the ratio between family income and poverty level by geographic area [24,25]. The poverty line in Brazil was set at BRL 178.00 per month per person [26]. The result of the PIR has been associated with the maternal education level, classifying families into a low, medium, or high SES.

The AHEMD-IS was used to verify the quantity and quality of affordances offered in the home environment. It was developed and validated in Brazil [13,14] and aims to evaluate the home environment according to the physical space, variety of stimuli, and the presence of gross and fine motor toys, classifying the home environment as less than adequate (LTA), moderately adequate (MA), adequate (A), and excellent (E) [13,14]. The AHEMD-IS has 35 questions and its responses are dichotomous (yes or no) or based on the Likert scale. The score of each dimension and the total score were calculated according to the age group. The total score ranges from 0 to 49 points. In this context, the score of each dimension is converted to the categories. The AHEMD-IS was scored directly by the caregiver (as a self-report) or by an interview with the caregiver. The choice of this questionnaire is justified by its ability to both quantify and qualify the affordances present in the home environment [27]. The instrument has reliable psychometric properties, with internal consistency measured using Cronbach’s alpha of 0.824 [0.781–0.862], inter-observer reliability (ICC 0.990), and intra-observer reliability (ICC 0.949) [13,14]. Maternal and paternal education were classified according to the International Standard Classification of Education (ISCED) and categorized as elementary school I and II (ISCED level < 3), high school completed (ISCED level 3–4), or higher (ISCED level 5–8) [28].

### 2.5. Procedures

After identifying eligible infants, their caregivers were contacted via telephone and a home visit was scheduled for those who were interested in study participation. During the home visit, information on the aspects of the infant’s birth, such as sex, gestational age, weight, length and head circumference at birth, and Apgar scores in the first and fifth minutes, was collected using the Adapted sociodemographic questionnaire [23]. Socioenvironmental factors such as maternal age, civil status, number of children and adults living in the house, the education levels of the father and mother, and questions regarding the affordances available in the home environment for motor development were collected through a direct interview and application of the AHEMD-IS [13,14].

### 2.6. Statistical Analyzes

Data were expressed as a mean and standard deviation or as a median and interquartile range (25–75%). Categorical data were presented in absolute and relative frequencies. Data normality was checked using the Kolmogorov–Smirnov test. Comparisons between the EG and GC were performed using the independent samples t-test or the Mann–Whitney test. The chi-square test was used to verify associations between sex and categories of the total score and dimensions of the AHEMD-IS (LTA, MA, A, and E).

A binary logistic regression model using the stepwise method, the selection analysis was performed using threshold, *p*-value < 0.05. It was also constructed to determine factors distinguishing environments classified as LTA between EG and GC groups. The goodness of fit was tested using the Hosmer–Lemeshow test, and the following parameters were reported: r^2^, odds ratio, and its 95% confidence interval. To mitigate type II error, effect sizes were computed for all tests.

Cohen’s d or r were calculated for intergroup analysis between parametric and nonparametric data, respectively, and interpreted as small (<0.50), medium (between 0.50 and 0.80), or large (>0.80) [29,30]. The strength of association in the chi-square test was verified using Cramer’s V and interpreted as low (<0.299), moderate (between 0.300 and 0.499), or high (>0.500) [31]. Inferential analysis was performed using the SPSS program (IBM Corp., Armonk, NY, USA) version 22. A two-tailed *p*-value < 0.05 was considered statistically significant for all analyses.

## 3. Results

### 3.1. Participants

Out of a total of 261 eligible infants 128 participated in the assessments. The reasons for non-inclusion of the infants, and details about the recruitment process are described in Figure 1.

The infants were divided into two balanced groups, as previously mentioned, 63 and 65 infants from 3 to 4 months old comprised the EG and CG groups, respectively. The characterization of the infants is described in both Table 1 and Table 2.

The infants and their families exposed to a low SES presented significant differences in length at birth, PIR, maternal age, civil status of the caregivers, number of children in the house, and maternal and paternal education. In contrast, they did not present significant differences in current age, i.e., at the moment of the assessment, gestational age, sex, birth weight, Apgar 1′, Apgar 5′, head circumference at birth, number of adults in the house, number of daughters and sons, i.e., sisters and brothers of the infant, and the role of the interviewed caregiver, i.e., each of the caregivers were mothers of the infants and were either on maternity leave or unemployed.

### 3.2. Home Environment of Infants Exposed to Poverty vs. Home Environment of Infants Who Are Not Exposed to Poverty, Comparison Group

Regarding the quantitative variables, the home environment of the EG presented a significantly lower provision of affordances for child development in the dimensions of gross motor toys (*p* < 0.0001 (Cohen’s r = 0.353); EG, median = 2.00 [1.00 to 3.00] vs. GC, median 3.00 [2.00 to 4.50]); fine motor toys (*p* = 0.0001 (Cohen’s r = 0.327); EG, median = 1.00 [0.00 to 2.00] vs. GC, median 2.00 [1.00 to 4.00]); and total score (*p* < 0.0001 (Cohen’s r = 0.377); EG, median = 15.00 [13.00 to 18.00] vs. GC, median 19.00 [16.00 to 22.00]). The results are detailed in Table 3.

### 3.3. Contextual Factors Associated with the Quantity and Quality of Affordances Less Than Adequate for Child Development

The binary logistic regression analysis showed a significant association (r^2^ = 0.828, *p* = 0.001) between maternal age and LTA category (*p* = 0.043, OR: 0.829 (0.692–0.994), with each year increase in maternal age the chance of the home environment scoring LTA decreases by 17.01%. Therefore, the analysis demonstrated that maternal age is considered a protective factor for the LTA category, regardless of the group. The regression analysis followed this model: > Y (Groups GE and GC—all MA) and X (mother’s age): r^2^ = 0.828, *p* = 0.001.

## 4. Discussion

The present study confirmed our hypotheses: (1) that the home environments of infants exposed to poverty presents LTA affordances for child development, mainly in the quantity and quality of gross and fine motor toys, leading to a lower total score compared to the home environments of infants who are not exposed to poverty; and (2) that maternal age is a protective factor for infants to present the LTA category in the total score of the AHEMD-IS, i.e., to obtain LTA affordances for child development in the home environment.

### 4.1. Home Environment of Infants Exposed to Poverty vs. Home Environment of Infants Who Are Not Exposed to Poverty

These results may be related to the fact that families experiencing poverty often possess limited knowledge regarding strategies to promote child development and may face constraints due to scarce economic resources. Certainly, economic resources are typically directed towards fulfilling basic necessities, such as food and housing, often relegating the purchasing of toys and materials to stimulate child development to a lower priority. Because the home environment significantly influences the stimulation of child growth and development, infants exposed to poverty may present deficits in domains such as motor skills [32], personal and social skills, problem resolution, and communication [33]. Consequently, the findings of this study emphasize the importance of public health professional networks offering clear and precise guidance to caregivers on how to stimulate the child development of infants exposed to poverty within the home environment, using alternatives which will be addressed in this discussion.

The dimensions of physical space and variety of stimulation did not show significant differences between the groups, although the EG displayed lower scores than the CG in both dimensions. This lack of significant difference in the EG, despite lower scores, can be attributed to the fact that the EG infants primarily reside in houses, whereas CG infants predominantly live in apartments, which often have more restricted spaces. However, the number of adults living in the infants’ houses in the EG is greater, which may contribute to lower quality of available physical space for the infants. The increased number of people in the house, as well as the low SES and caregiver’s education, may contribute to domestic chaos [34]. Domestic chaos is characterized by a disorganized noisy environment, a lack of family routine, a fast-paced life, and the absence of planning and structure when performing the activities of daily living [35].

To improve the affordances provided in the home environment’s physical space dimension for infants, it is recommended to encourage and involve all adults residing in the infant’s household in daily care and the provision of developmental stimuli. This inclusive participation has been proposed to establish protective factors for child development. It was observed that in the present study, only the infants’ mothers were available to respond to the AHEMD-IS, in addition to learning about stimuli for child development through the researchers’ guidelines. Therefore, these mothers may experience challenges in managing infant care alongside their personal demands. Previous studies have shown a positive association between weight, cognitive, fine-motor and socio-emotional development during early childhood and the involvement of extended family members, such as grandmothers, in the infant’s routine [36]. Given that parents and close family members serve as primary caregivers in the initial years of life, family-centered care emerges as extremely important [37].

Regarding the variety of stimulation dimension, it is hypothesized that the presence of a greater number of children in the household within the EG may have contributed to the relatively modest disparity between the home environments of the EG and CG. It was observed that the home environment of the EG exhibited a higher count of children when compared to the CG. Children are important factors in offering a variety of stimulation and fostering child development, as they offer challenges to infants and are more available for interaction and the encouragement of play. The infant’s coexistence with other children in the home environment not only stimulates child development but also optimizes the motor learning process. This is attributed to the infant’s capacity to learn from the actions performed by other children, due to neural mirroring mechanisms [38], a phenomenon present even in newborns [39,40] which becomes increasingly acute during the first year of life [41].

In relation to gross and fine motor toys, they are crucial for infants to be able to perform activities such as rolling, walking, crawling, reaching, manipulating, exploring an object, pinching, and consequently preparing for future motor skills, for example, grasping and writing. Environmental enrichment and effective parenting practices will enhance the optimization of affordances for the motor development present in the home environment, regardless of SES [15,42].

The AHEMD-IS interview provided not only the assessment of the home environment, but also the possibility of guiding mothers on how to improve the provision of adequate stimuli for child development, such as selecting toys and activities suitable for their infants within the environment home. The guidelines were based on the principle that mothers could offer adequate affordances for their children’s motor development without significant financial expenditure. Researchers instructed mothers on how to produce toys with recyclable materials and offer motor development stimuli during infant care activities. The suggested practices included crafting rattles from plastic and paper packaging and using cardboard boxes to engage infants in fine motor activities such as kneading, tearing, and squeezing. Additionally, guidance on sensory stimuli with different textures using bath and kitchen sponges, cotton, towels, paper packaging, as well as the importance of dyad interaction, for example, at bath time, and teaching them about their body parts was provided. Furthermore, mothers were encouraged to offer their infants the possibility to experience various postures, such as prone, sitting and/or free to move on the floor, in corroboration with the findings of Cunha et al. [43].

We highlight that although toys made from recyclable materials are an excellent strategy for families exposed to poverty, the use of these toys produced at home must be supervised by adults due to the inherent difficulty in guaranteeing their safety. Given the age of the infants in our study, caution should be taken when using such toys because this developmental stage involves infants placing objects into their mouths, presenting a potential risk for accidents or asphyxiation with non-industrialized toys/materials, just as there is a risk with industrialized materials that are not suitable for the infant’s age.

Therefore, the present study suggests the provision of basic kits of varied and age-appropriate toys as part of the routine monitoring of infant growth and development, especially infants from families exposed to poverty. This is a pragmatic and low-cost measure that could contribute to environmental enrichment, and consequently, enhance overall child development. The study by Cunha et al. [43] shows the feasibility of stimulating child development in families exposed to poverty by the application of a simple protocol of child development stimuli that uses only three toys of different shapes, suitable for the infant’s age, such as rattles, stacking pots, and a plastic baby book.

Although the variables of length, weight, and head circumference at birth are not related to the affordances of the home environment for child development, interestingly, it was noted that infants exposed to poverty had a significantly lower length at birth when compared to infants who are not exposed to poverty. Although the infants did not present a significant difference in head circumferences and birth weights, it is crucial to underscore the importance of monitoring pregnancy through prenatal consultations of families exposed to poverty.

Families exposed to poverty may have difficulties in performing follow-ups for prenatal consultations and maintaining healthy pregnancies, perhaps due to factors such as a higher proportion of teenage mothers facing unplanned pregnancies [44], or inadequate nutrition due to lack of economic resources [45]. Significantly, the maternal age within EG was lower than that of the CG. Therefore, we emphasize that it is essential to have public policies that support maternal childcare, encompassing comprehensive pre- and post-natal follow-ups for mothers and infants exposed to poverty within the primary care network [46]. This difference in maternal age between groups can be attributed to various contributing factions, including access to quality education, gender equality, contraceptive methods, participation in the labor market, and availability of assisted reproductive technology. Families with access to these resources prioritize pursuing higher educational and financial achievements before building families [47,48].

### 4.2. Contextual Factors Associated with the Quantity and Quality of Affordances Less Than Adequate for Child Development

We observed that maternal age may be a protective factor for the home environment to present adequate quantity and quality of affordances for neuropsychomotor development. More mature women may be more prepared to dedicate themselves to infant care, providing affordances through a structured home environment, varied postural stimulation, and diverse toys. Previous research has showed that children born to more experienced mothers tend to perform better academically and behaviorally [49]. Notably, in our research, with each additional year of maternal age the chance of the home environment presenting LTA affordances for child development decreases by 17.01%, regardless of the group.

### 4.3. Strengths and Limitations

The present study provides relevant scientific evidence into the affordances offered in the home environments of infants exposed to poverty using the AHEMD-IS. However, the AHEMD-IS was not subsequently applied to assess whether the home guidelines had an effect on the quality of the affordances present in the home environment over the time. It is suggested that future clinical trials be performed to evaluate the effectiveness of home guidance protocols, early intervention, and optimization of the home environment. Clinical trials should be designed according to the low-cost, easy-to-apply biopsychosocial model.

## 5. Conclusions

The home environments of infants exposed to poverty presented suboptimal affordances for motor development, mainly in the dimensions of gross motor toys, fine motor toys and, consequently, the total score. In contrast, a positive correlation was observed between higher maternal ages and better results regarding the quantity and quality of affordances for child development present in the home environment. In summary, given the important impact of environmental factors on this functionality, we suggest that greater attention should be given to specialized individualized guidance programs centered on the biopsychosocial model, particularly tailored for families exposed to poverty.

## Figures and Tables

**Figure 1 children-10-01932-f001:**
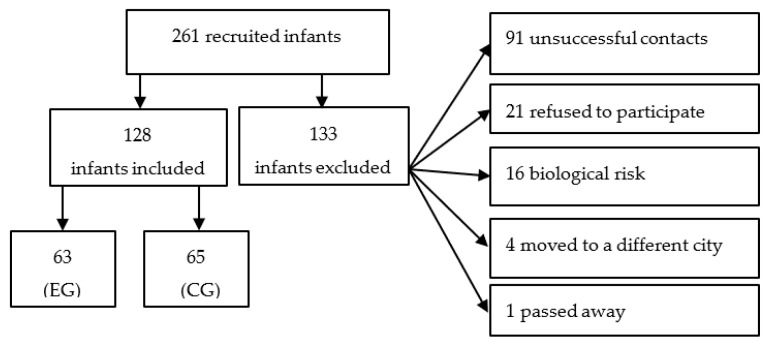
Infant recruitment flowchart. Legend: EG, exposed group; CG, comparison group. Source: own authorship.

**Table 1 children-10-01932-t001:** Characterization of aspects at birth, socio-environmental and socio-economic factors of the sample, continuous variables.

Characteristics	EG (*n* = 63)	CG (*n* = 65)	*p*-Value
Current age (in days): mean (SD)	106.55 ± 14.22	108.88 ± 14.64	*p* = 0.809 ^b^ (Cohen’s r = 0.054)
Gestational age (in weeks): mean (SD)	39.04 ± 1.96	38.88 ± 1.21	*p* = 0.075 ^b^ (Cohen’s r = 0.160)
Birth weight (in grams): mean (SD)	3229.00 ± 354.10	3 375.00 ± 478.70	*p* = 0.062 ^a^ (Cohen’s d = 0.346)
Apgar 1′ mean (SD)	8.65 ± 0.98	8.71 ± 1.10	*p* = 0.425 ^b^ (Cohen’s r = 0.066)
Apgar 5′ mean (SD))	9.80 ± 0.41	10.60 ± 5.45	*p* = 0.392 ^b^ (Cohen’s r = 0.050)
Length at birth (in centimeters) mean (SD)	46.65 ± 3.96	48.57 ± 2.40	***p* = 0.033 ^b^** **(Cohen’s r = 0.157)**
Head circumference at birth (in centimeters) mean (SD)	35.51 ± 3.83	34.80 ± 1.63	*p* = 0.733 ^b^ (Cohen’s r = 0.068)
PIR mean (SD)	1.21 ± 0.47	5.94 ± 7.22	***p* < 0.01 ^b^** **(Cohen’s r = 0.351)**
Maternal age (in years) mean (SD)	24.08 ± 5.99	31.63 ± 6.89	***p* < 0.01 ^b^** **(Cohen’s r = 0.501)**
Number of children in the house median (min–max)	2 (1–6)	1 (1–2)	***p* < 0.01 ^b^** **(Cohen’s r = 0.290)**
Number of daughter/son median (min–max)	2 (1–6)	1 (1–4)	*p* = 0.083 ^b^ (Cohen’s r = 0.142)
Number of adults in the house median (min–max)	2 (1–7)	2 (2–2)	*p* = 0.355 ^b^ (Cohen’s r = 0.070)
Number of rooms in the residence: median (min–max)	2 (1–3)	2 (1–5)	*p* = 0.096 ^b^ (Cohen’s r = 0.130)

Legend: **Bold**, variables that showed significant difference; min, minimum; max, maximum; SD, standard deviation; *n*, absolute frequency; %, relative frequency; PIR, poverty income ratio; EG, exposed group; CG, comparison group; ^a^, independent *t*-test; ^b^, Mann–Whitney test. Source: own authorship.

**Table 2 children-10-01932-t002:** Characterization of aspects at birth, socio-environmental and socio-economic factors of the sample, categorical variables.

Characteristics	Categories	GE (*n* = 63)	CG (*n* = 65)	*p*-Value
Sex *n* (%)	Male	33 (52.38)	35 (53.84)	*p* = 0.869 ^a^ (Cohen’s r = 0.014)
	Female	30 (47.61)	30 (46.15)	
Civil status *n* (%)	Stable union	45 (71.42)	61 (93.84)	***p* < 0.01 ^a^** **(Cohen’s r = 0.311)**
	Single	18 (28.57)	4 (6.15)	
	Divorced	0 (0)	0 (0)	
Maternal education *n* (%)	Completed/incompleted Fundamental school	63 (100.00)	0 (0)	***p* < 0.01 ^a^** **(Cohen’s r = 0.735)**
	Incomplete high school	0 (0)	10 (15.36)	
	High school	0 (0)	38 (58.45)	
	College or postgraduation	0 (0)	17 (26.15)	
	Not informed	0 (0)	0 (0)	
Paternal education *n* (%)	Completed/incompleted Fundamental school	40 (63.48)	21 (32.30)	***p* < 0.01 ^a^** **(Cohen’s r = 0.457)**
	Incomplete high school	9 (14.28)	28 (43.07)	
	High school	(0)	11 (16.91)	
	College or postgraduation	0(0)	0(0)	
	Not informed	14 (22.22)	5 (7.89)	
The role of the interviewed caregiver *n* (%)	Mother	63 (100)	65 (100)	*p* = 1.000 ^a^ (Cohen’s r = 0)
	Father	0 (0)	65 (100)	
	Other	0 (0)	65 (100)	

Legend: **Bold**, variables that showed significant difference; min, minimum; max, maximum; SD, standard deviation; *n*, absolute frequency; %, relative frequency; PIR, poverty income ratio; EG, exposed group; CG, comparison group; ^a^ Mann–Whitney test. Source: own authorship.

**Table 3 children-10-01932-t003:** Qualitative comparison regarding the affordances presented in the home environments of the exposed group vs. comparison group, according to the data collected using the AHEMD-IS.

Domains	X^2^ (df = 3, *n* = 128)	*p* Value	Cramer’s V
Physical space	2.333	0.506	0.135
Variety of stimulation	2.865	0.413	0.150
Gross motor toys	15.87	*p* < 0.001 *	0.350
Fine motor toys	8.419	*p* = 0.038 *	0.256
Total score	17.306	*p* < 0.001 *	0.368

Legend: X^2^, chi-square; df, degrees of freedom, * presented significant differences between the groups. Source: own authorship.

## Data Availability

The data presented in this study are available on request from the corresponding author. The data are not publicly available due to privacy restrictions.

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
