# Peer review of "Association between Contextual Factors and Affordances in the Home Environment of Infants Exposed to Poverty"

_children, 2023, doi:10.3390/children10121932_

Round 1

Reviewer 1 Report

Comments and Suggestions for Authors

Missing from the description of the methodology is the method of selecting probands and how to contact them. Without a more detailed description, it is not clear whether the sample was locally specific or influenced by the selection intention.

In chapter 2.3, line 102, I assume missing months of age. The description is unclear: ", infants of 101 both sexes at 3 and 4 years of age were selected for the study" vs. "full-term infants of both sexes were selected at 3, 4, 5, 6, 7 and 8 months old" (line 105)

It would be appropriate to add the language of the instrument used (AHEMD-IS), including the method of standardization for this language.

Figure 1/Table 1 - the abbreviation CG and EG was used once and GC and GE a second time.

Reviewer 2 Report

Comments and Suggestions for Authors

Title and Abstract are well-written and provide the basic information about study.

Introduction: It cite relevant literature and provides scope of study and significance.

Method: Adequately provide information about study methods.

Results: Table 1 needs to be revised as it is providing wide range of statistics, it not clear whether it is showing n or % or means and S.D.. Moreover, the data related to education has categories but the way it is presented is making it too much complicated to understand. Its preferable to distinguish between continuous and categorical variables and then report them accordingly in separate tables. Similarly table 2 has errors, report only two decimal points for chi-square findings plus at some points comma is used and other points decimal point. It is important to review the result section of the manuscript and improve the descriptive and tabular findings of the study to ensure clarity in terms of study findings and its interpretation.

Discussion: It seems appropriate literature and explanations are used to support study findings and their interpretation.

Comments on the Quality of English Language

Minor editing is needed to improve the sentence structure in the discussion section. E.g. "It is suggested that clinical trials be carried out to test the effectiveness of 329 home guidance protocols, early intervention and optimization of the home environment, 330 according to the low-cost, easy-to-apply biopsychosocial model." This is very long sentence and the reader feel lost what is mean by applying biopsychosocial model etc. 

Results sections need more improvement to enhance the clarity. 

Reviewer 3 Report

Comments and Suggestions for Authors

This paper has potential but needs much improvement. I have attached hand-written corrections, which will help. For whatever cannot be read, please send a snapshot and I will clarify. Beyond corrections, there are substantive issues. 1. The abstract should define terms like AHEMD-IS and MQA. “fine” is missing “motor toys.” 2. On page 2, I note that the literature review on home affordances is sparse. There is a whole set of studies using the HOME questionnaire that should be cited. The literature review should indicate what id lacking in prior studies, setting the stage for the hypotheses and design. On page 3, The N is not given. Line 102 is confusing, mentioning years when only months are involved. Line 105 needs to specify the number of infants at each age. For instruments, the AHEMD-IS administration who needs description of the procedure/ instructions, who scored it, etc. The five dimensions and their abbreviations need mention here . Were cut scores used to categorize the infants? On page 5, the 2 groups arrived at near identical N’s. Was that by chance, or were there predetermined N’s that were used to select infants? On page 5, for Table 1, you use the results in the discussion, but give a summery here. Another statistical approach would be to see if the demographics that differed in the groups were used as covariates in the regression. Some sentences were too long and not well written. Split them and check their semantics (e.g., Lines 206-208 on page 6, Lines 324-331 on page 8). Once rewritten, [perhaps use a writing service to check the writing in English. Conclusion: major revision required.

Comments on the Quality of English Language

This paper has potential but needs much improvement. I have attached hand-written corrections, which will help. For whatever cannot be read, please send a snapshot and I will clarify. Beyond corrections, there are substantive issues. 1. The abstract should define terms like AHEMD-IS and MQA. “fine” is missing “motor toys.” 2. On page 2, I note that the literature review on home affordances is sparse. There is a whole set of studies using the HOME questionnaire that should be cited. The literature review should indicate what id lacking in prior studies, setting the stage for the hypotheses and design. On page 3, The N is not given. Line 102 is confusing, mentioning years when only months are involved. Line 105 needs to specify the number of infants at each age. For instruments, the AHEMD-IS administration who needs description of the procedure/ instructions, who scored it, etc. The five dimensions and their abbreviations need mention here . Were cut scores used to categorize the infants? On page 5, the 2 groups arrived at near identical N’s. Was that by chance, or were there predetermined N’s that were used to select infants? On page 5, for Table 1, you use the results in the discussion, but give a summery here. Another statistical approach would be to see if the demographics that differed in the groups were used as covariates in the regression. Some sentences were too long and not well written. Split them and check their semantics (e.g., Lines 206-208 on page 6, Lines 324-331 on page 8). Once rewritten, [perhaps use a writing service to check the writing in English. Conclusion: major revision required.

Round 2

Reviewer 3 Report

Comments and Suggestions for Authors Dear Author(s)   This version is written better. I have added minor writing corrections to your paper in hand writing to consider further. There is only one substantive comment, to check the HOME measure and indicate how the one you chose better fits your hypotheses, assuming that is the case. I indicated where to address this in the handwritten corrections. Overall, these are minor corrections.

Comments on the Quality of English Language See above.
